# H3AE: HIGH COMPRESSION, HIGH SPEED, AND HIGH QUALITY AUTOENCODER FOR VIDEO DIFFUSION MODELS

## ABSTRACT

Autoencoder (AE) is the key to the success of latent diffusion models for image and video generation, reducing the denoising resolution and improving efficiency. However, the power of AE has long been underexplored in terms of network design, compression ratio, and training strategy. In this work, we systematically examine the architecture design choices and optimize the computation distribution to obtain a series of efficient and high-compression video AEs that can decode in real time even on mobile devices. We also propose an omni-training objective to unify the design of plain Autoencoder and image-conditioned I2V VAE, achieving multifunctionality in a single VAE network but with enhanced quality. In addition, we propose a novel latent consistency loss that provides stable improvements in reconstruction quality. Latent consistency loss outperforms prior auxiliary losses including LPIPS, GAN and DWT in terms of both quality improvements and simplicity. H3AE achieves ultra-high compression ratios and real-time decoding speed on GPUs and mobile devices, and outperforms prior arts in terms of reconstruction metrics by a large margin. We finally validate our AE by training a DiT on its latent space and demonstrate fast, high-quality text-to-video generation capability.

## 1 INTRODUCTION

Over the past year, diffusion-based generative models have achieved remarkable progress, extending the success of text-to-image (T2I) models (Rombach et al., 2022a; Podell et al., 2023; Esser et al., 2024; Black Forest Labs, 2023; Betker et al., 2023; Baldridge et al., 2024) to the more complex text-to-video (T2V) generation (Brooks et al., 2024; Zheng et al., 2024; Polyak et al., 2024; Ma et al., 2025; Wan et al., 2025). Recent advances from both industry (Polyak et al., 2024; Veo-Team et al., 2024; Brooks et al., 2024) and academia (Yang et al., 2024; Zheng et al., 2024; Team, 2024a) enable the creation of cinematic videos from text prompts (Ma et al., 2025; Veo-Team et al., 2024) or image inputs (Blattmann et al., 2023; Polyak et al., 2024). These foundation models also unlock new applications such as video editing (Jeong et al., 2024; Liang et al., 2023) and 3D generation (Voleti et al., 2024; Kwak et al., 2024).

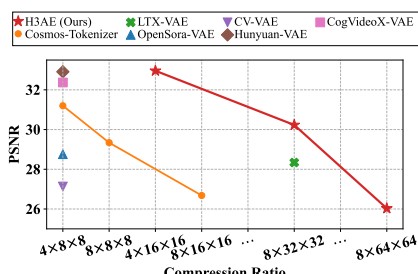

Figure 1: Compression ratio is in $log_2$ scale. H3AE achieves a better compression–PSNR trade-off and is faster and more parameter-efficient. Refer Tab. 3 for more benchmarks.

The key driver behind this success is Latent Diffusion Modeling (LDM) (Rombach et al., 2022a; Liu et al., 2022). Unlike pixel diffusion models (Menapace et al., 2024; Saharia et al., 2022; Ho et al., 2022; DeepFloyd, 2023) which learn the diffusion mapping from noise to raw pixel space, LDMs learn the diffusion process in a compressed latent representation provided by a Variational Autoencoder (VAE) (Kingma & Welling, 2013; Yu et al., 2023; Agarwal et al., 2025), yielding dramatic improvements in both training and inference efficiency. The effectiveness of VAE critically

depends on three main factors: (i) *Reconstruction quality*, imperfect reconstruction introduces artifacts into the generation pipeline. (ii) *Compression ratio*, which directly determines the efficiency of the denoising process: compressing latent resolution by a factor of $O(n)$ yields up to $O(n^2)$ computational savings for transformer-based denoisers. (iii) *Architectural efficiency*, especially for decoding, which has emerged as a speed bottleneck after recent advances in reducing denoiser complexity and denoising steps (Li et al., 2023; Hu et al., 2024; Zhao et al., 2024b; Wu et al., 2024; Kulikov et al., 2023; Xu et al., 2024; Zhang et al., 2024). Despite their centrality, VAEs in diffusion pipelines have received less systematic study compared to denoisers. Existing video VAEs (Agarwal et al., 2025; HaCohen et al., 2024) explored design choices in an ad-hoc manner, such as adding input pachifications or bringing/discarding auxiliary losses, but leave open key questions about the compression-quality trade-off, the architectural balance between speed and fidelity, and effective training strategies. In this work, we introduce H3AE, a holistic framework that addresses the two principal domains for video VAE optimization:

**From the efficiency perspective**, we explore VAE designs in terms of: **(a)** *Decoding Efficiency (VAE architecture)*: We design a compact, low-latency decoder tailored for video autoencoding. Rather than adopting causal 3D convolutions throughout, we structure the decoder in disentangled stages using 2D Conv, 3D Conv and causal attentions, promoting efficiency and enabling sliced decoding for high resolution stages. Through profiling and ablation, we distribute computation across layers to balance speed, memory, and reconstruction fidelity, resulting in a decoding backbone that is fast enough for real-time decoding or even mobile usage while maintaining strong reconstruction. **(b)** *Denoising Efficiency (Compression Ratio)*: Thanks to our efficient but powerful architecture, we push the compression ratio further than prior VAEs while still preserving high-quality reconstructions, as in Fig. 1. This more aggressive compression reduces the token count fed into the diffusion model, thereby accelerating denoising and lowering memory costs. To validate that our VAE still supports effective diffusion, we train a video DiT model on its latent space and demonstrate high-quality, fast video generation—evidence of "diffusability".

**On the training algorithm side**, we propose two methods to improve quality: **(a)** *Omni-Objective Training:* Recently, Reducio (Tian et al., 2024) proposed a VAE design for image-to-video generation by leaking the first frame to the decoder to improve reconstruction quality with a higher compression ratio compared to T2V. To leverage this advantage and still support a single VAE for T2V and I2V generations, we propose a novel design that unifies decoding irrespective of the presence of image condition. We randomly feed the hierarchical features of the first frame as a condition to the decoder during training, such that our VAE can optionally act as an I2V VAE with improved quality. Surprisingly, this training strategy improves reconstruction even without image conditioning (plain T2V setting) due to the effect of training augmentation. **(b)** *Latent Consistency Loss:* Existing work (Yang et al., 2024; Kong et al., 2024; HaCohen et al., 2024; Wang et al., 2025) mainly use L1 reconstruction loss, KL regularization, and auxiliary losses (LPIPS, GAN, DWT) to train the VAE. We empirically show that the quality gain from these pre-defined auxiliary losses are limited, and they are at risk of introducing checkerboard artifacts. We propose a new training strategy where we first train only with reconstruction loss and KL regularization for faster convergence. Then, we utilize a novel latent consistency loss to further finetune the model by re-encoding the reconstructed video to obtain the fake posterior and optimize its KL divergence against the former posterior encoded from the ground truth video. In our experiments, latent consistency loss is simple to integrate, stable under training, and consistently improves reconstruction fidelity, outperforming previous auxiliary losses.

Our contributions can be summarized as follows,

- We systematically explore the design space of video VAEs (normalization, upsampling, temporal vs spatial factorization, causal attention, etc.) and propose a new architecture that realizes strong reconstruction, high compression, and low-latency decoding.

- We propose an omni-objective training method by randomly injecting first-frame hierarchical features to the decoder during training, enabling one VAE to support both T2V and I2V settings, and improving reconstruction even in the unconditioned T2V setting due to training-time augmentation.

- We propose a novel latent consistency loss to further boost the reconstruction quality, outperforming prior auxiliary losses such as LPIPS, GAN and DWT. Latent consistency loss is stable and simple to integrate into VAE training.

- We validate H3AE by training a Video DiT in its highly compressed latent space, demonstrating nice diffusability and fast generation speed.

## 2 Related Work

**Latent Diffusion Models.** Diffusion models (Vahdat et al., 2021; Rombach et al., 2022a; Song et al., 2021; Ho et al., 2020; Nichol & Dhariwal, 2021; Karras et al., 2022) have emerged as the leading framework for generative modeling, surpassing earlier approaches such as GANs (Goodfellow et al., 2014; Karras et al., 2019) and VAEs (Kingma & Welling, 2013). These models generate stunning visuals by progressively denoising noise into meaningful context, guided by inputs such as text prompts or images. Early text-to-image (T2I) (Saharia et al., 2022; DeepFloyd, 2023) and text-to-video (T2V) (Menapace et al., 2024; Ho et al., 2022) diffusion models operated directly in pixel space, but this was computationally prohibitive due to the need for very deep networks. Latent Diffusion Models (LDMs) (Rombach et al., 2022a; Blattmann et al., 2023) addressed this issue by first compressing pixels into a semantically rich latent space via an autoencoder, and then applying the diffusion process in that space. This dramatically reduced computational costs while preserving generative quality. Today, most state-of-the-art image (Podell et al., 2023; Esser et al., 2024; Black Forest Labs, 2023; Kag et al., 2024; Gao et al., 2024; Team, 2024b; Liu et al., 2024; Li et al., 2023; Hu et al., 2024) and video (Blattmann et al., 2023; Brooks et al., 2024; Zheng et al., 2024; Menapace et al., 2024; Polyak et al., 2024; Team, 2024a; HaCohen et al., 2024; Yang et al., 2024; Wu et al., 2024) models adopt this paradigm, achieving visually compelling and semantically aligned outputs.

**Video Autoencoders.** Since autoencoders define the latent space of LDMs, their design directly impacts both quality and efficiency. Early video LDMs (Blattmann et al., 2023; Guo et al., 2023) reused image-based $8\times8$ spatial autoencoders, which failed to compress temporal redundancy. To enable longer videos, later works introduced spatio-temporal compression schemes, such as $4\times8\times8$ (Yang et al., 2024; Zheng et al., 2024; Zhou et al., 2024; Kong et al., 2024), and causal temporal structures (Yu et al., 2023). More recent models push compression further, exploring $8\times8\times8$ (Polyak et al., 2024; Agarwal et al., 2025; Tang et al., 2024), $8\times16\times16$ (Ma et al., 2025), or even $8\times32\times32$ (HaCohen et al., 2024). While extreme compression unlocks faster diffusion (Xie et al., 2024; Chen et al., 2025b; HaCohen et al., 2024), it risks poor reconstructions and requires careful validation during diffusion training (Skorokhodov et al., 2025).

**Architecture.** Most video autoencoders rely on convolutional backbones (Yang et al., 2024; Sadat et al., 2024; Zhao et al., 2024a), but alternative designs are emerging. Wavelet-based methods (Graps, 1995; Lin et al., 2024; Agarwal et al., 2025; Li et al., 2024) offer efficient handling of high-resolution data, while transformer-based approaches (Esteves et al., 2024; Chen et al., 2025b; Hansen-Estruch et al., 2025; Yu et al., 2024; Chen et al., 2025a) leverage tokenization for scalable latent representations. Recent 1D image tokenizers (Yu et al., 2024; Chen et al., 2025a) push compression to the extreme (up to $2048\times$) by adopting ViT-like (Dosovitskiy et al., 2020) autoencoding backbones, hinting at new possibilities for video representation learning.

## 3 Method

### 3.1 VAE Architecture

#### 3.1.1 Micro Design

Inspired by recent video VAEs (Yang et al., 2024; Agarwal et al., 2025; HaCohen et al., 2024), we start from a plain autoencoder backbone built with 3D Causal Convolutions, and explore the following design choices to construct a powerful yet efficient architecture.

**Spatial Stage**. Applying Conv3D at every stage of a video autoencoder can result in excessive memory consumption, especially in high-resolution stages, thereby hindering its efficient mobile deployment. To mitigate this issue, we disentangle the autoencoder structure by using 2D spatial convolutions in high-resolution stages, as illustrated in Fig. 2. This disentanglement reduces computation and peak memory for high-resolution stages and, in addition, allows frame-by-frame chunked inference. Notably, disentangling the high-resolution stage with Conv2D only has a negligible impact on the reconstruction quality compared to Conv3D, as shown in Tab. 1, while the

Table 1: **AE Design Ablation.** We ablate on various design choices for autoencoder architecture. The decoding latencies are benchmarked on iPhone 16 PM with 17 frames $512 \times 512$ resolution output and the reconstruction quality evaluations are conducted on DAVIS dataset.

| Spatial | Upsample | $|z|$ | Norm | Attention | Params (M) | FPS@$512 \times 512$ | PSNR | SSIM | rFVD |
|---------|----------|-------|------|-----------|------------|----------------------|------|------|------|
| **2D** | **PixelShuffle** | **256** | **PN** | ✓ | **196** | **38.1** | **29.48** | **0.8280** | **147.1** |
| 3D | PixelShuffle | 256 | PN | ✓ | 189 | ✗ | 29.53 | 0.8255 | 126.4 |
| Patchify | PixelShuffle | 256 | PN | ✓ | 194 | 43.5 | 28.46 | 0.7990 | 195.3 |
| 2D | Interpolation | 256 | PN | ✓ | 169 | 40.1 | 24.78 | 0.7277 | 995.8 |
| 2D | PixelShuffle | 128 | PN | ✓ | 195 | 38.3 | 28.34 | 0.7963 | 300.9 |
| 2D | PixelShuffle | 256 | GN | ✓ | 197 | 37.9 | 29.43 | 0.8242 | 151.7 |
| 2D | PixelShuffle | 256 | LN | ✓ | 197 | 27.5 | 29.49 | 0.8288 | 144.8 |
| 2D | PixelShuffle | 256 | PN | ✗ | 316 | 41.7 | 28.59 | 0.8052 | 246.7 |

non-parameterized patchify introduced by LTX (HaCohen et al., 2024) demonstrates degraded performance.

**3D Causal Attention**. We introduce 3D Causal Attention as an alternative foundation block in our VAE. 3D Causal Attention has a global spatial receptive field, while only attends to present and previous frames in the temporal dimension. Tokens are flattened from $\mathbb{R}^{T \times H \times W \times C}$ to $\mathbb{R}^{(T \cdot H \cdot W) \times C}$ before being processed by the attention mechanism. To enforce causality, a block causal mask is applied to the attention score, where each block has the size of $L_H \times L_W$, indicating that the tokens are within a single frame, as illustrated in Fig. 2 (right). The mask ensures that video tokens from future frames remain inaccessible to tokens from earlier frames, which preserves temporal consistency and enables arbitrary-length video processing. We adopt Rotary Positional Embeddings (RoPE) to encode relative positional information by applying learned rotations directly in the query–key inner product space, which supports better extrapolation and long-context modeling compared to absolute positional embeddings (Su et al., 2024). Additionally, we employ QK-norm normalization (Henry et al., 2020), which normalizes the query and key vectors before attention computation to stabilize training and prevent attention saturation, especially under high-compression and long-sequence settings. As in Tab. 1, using Causal 3D Attention reduces 38% parameters, but achieves 0.9 higher PSNR.

**Upsampling**. Besides traditional interpolation methods, PixelShuffle has become a popular trend in Autoencoders (Chen et al., 2025b). We observe that PixelShuffle gives much better reconstruction quality than interpolation, with negligible overhead in parameters and latency, as in Tab. 1.

**Latent Channels**. Prior continuous variational autoencoders typically adopt a small number of latent channels, such as 4-channel in stable diffusion (Rombach et al., 2022b). However, recent work has shown that using more latent channels not only improves reconstruction quality, but also offers better semantic completeness and aesthetic quality for diffusion models (Esser et al., 2024; Hong et al., 2022; Hu et al., 2024). We explore larger number of latent channels for our VAE (*i.e.* 128-channel and 256-channel) in Tab. 1. We find that both 128-channel and 256-channel give reasonable reconstruction performance, and the 256-channel one achieves relatively higher PSNR. We report video diffusion results for both settings.

**Normalization Layers**. Though static BatchNorm is foldable during the inference phase thus most efficient, we find that it fails to generalize to different resolutions for VAE reconstruction. We explore dynamic norms including PixelNorm (Karras et al., 2019) (PN), LayerNorm (LN) and GroupNorm (GN) in this work. Among them, LN is the default choice in transformer-based models, while GN is popular in CNN-based generative models (Hong et al., 2022; Yang et al., 2024). We find that PN, LN, and GN have similar reconstruction performance, as in Tab. 1. Note that PN is the simplest without learnable parameters, and in addition, LN and GN require reshape operations to process 5D video data, which are less efficient on mobile devices. As a result, we choose PN as our normalization method.

### 3.1.2 MACRO ARCHITECTURE

Wrapping up the design, we divide both the encoder and the decoder into two stages. The high-resolution stage utilizes spatial Conv2D for sliced inference, the spatial-temporal stage uses 3D Causal convolution blocks as well as 3D Causal Attention to model temporal dependency and capture global receptive field, as in Fig. 2. Note that even though multiple downsampling (upsampling) sub-stages are involved in the spatial-temporal stage, we only distribute 3D Causal Attention in

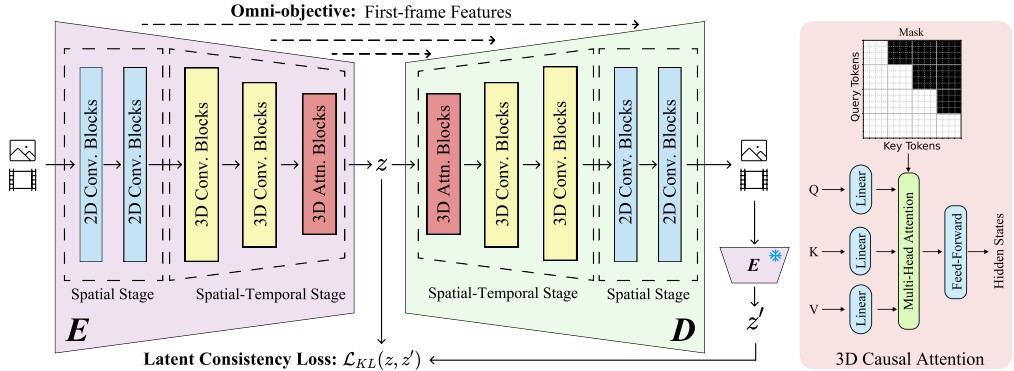

Figure 2: **Overview of H3AE architecture, omni-objective training, and Latent Consistency Loss**. When computing $z'$ in Eq. (3), the encoder weights remain frozen. For omni-objective training, we randomly pass the hierarchical features of the first frame from the encoder to the decoder, and use addition by default for feature fusion. As in the right, a block-shaped causal mask is applied to the 3D Transformer to enforce the causality of the attention mechanism, ensuring proper temporal dependencies in the generated representations.

the bottleneck position with the highest compression ratio. This straightforward design aligns the computation and memory complexity of the 3D Attentions in VAE with those in the DiT denoiser, ensuring that the entire diffusion pipeline is runnable on resource-constrained devices. For instance, our $8 \times 32 \times 32$ H3AE has only $1/32$ tokens in the bottleneck and latent stage compared to the CogVideoX $4 \times 8 \times 8$ VAE, thus the 3D Attentions in our H3AE and potential DiT denoiser enjoy more than $1000\times$ FLOPs reduction compared to CogVideoX DiT, significantly improving computational efficiency for video generation.

Besides architectural design, it is also crucial to properly distribute network parameters and computations to achieve a better Pareto curve of performance and efficiency. We investigate the depth and width scaling property of our VAE by directly profiling on the mobile device (iPhone 16 PM). We intuitively set the maximum network width based on the memory bound, and set a real-time target for mobile inference. i.e., decoding a $17 \times 512 \times 512$ video clip within 0.5s, which is equivalently more than 30 FPS. We scale the network by shrinking the width but

Table 2: Scaling of network backbone. The $1\times$ backbone is constructed under *real-time* constraint while maximizing the width to fit mobile memory.

| Width | Params (M) | Latency (s) | PSNR |
|-------|-----------|-------------|-------|
| $0.5\times$ | 233.0 | 0.45 | 27.02 |
| $0.75\times$ | 206.1 | 0.46 | 29.11 |
| $1\times$ | 196.5 | 0.45 | 29.48 |

increasing the depth to maintain this real-time target, and explore reconstruction performance, as in Tab. 2. We find that in this scope, a wide but shallow VAE achieves the best reconstruction quality. All of our H3AEs are then constructed based on depth-width ratios of the $1\times$ variant in Tab. 2.

## 3.2 OMNI-AE FOR BOTH T2V AND I2V

Image-to-video (I2V) generation is a prominent application in video generation (Yang et al., 2024; Zheng et al., 2024; Tian et al., 2024), aiming to synthesize video sequences conditioned with a user-specified input image (typically the first frame). This task is crucial for various applications such as video prediction and animation. Tian et al. (Tian et al., 2024) propose to construct an image-conditioned VAE to utilize this additional information, achieving better reconstruction quality in high compression settings. Although using the image condition benefits quality, the dedicated nature of I2V-VAE prohibits its wide deployment. Considering the cost of adapting the denoiser to a new VAE when swapping from T2V to I2V, most video diffusion works (Blattmann et al., 2023; Hong et al., 2022; Polyak et al., 2024) still use plain VAEs for simplicity.

In this work, we propose a simple yet effective multifunctional VAE that works for both plain T2V and conditioned I2V settings. Specifically for the decoder ($\hat{x} \leftarrow \text{Decoder}(z)$), we propose an omni-objective training strategy, where we take the hierarchical features ($e_i$) of the first frame from the

encoder and feed them to the decoder as conditions with probability $p$.

$$z_{i+1} = \begin{cases} \text{DecoderBlock}_i(z_i), & \text{with probability } 1-p, \\ \text{DecoderBlock}_i\big((z_i + e_{N-i})/2\big), & \text{with probability } p, \end{cases} \quad (1)$$

where $z_i$ and $e_{N-i}$ are the decoder and encoder features at symmetric positions assuming a total of $N$ decoder blocks, $z_0$ is the latent and $z_N$ is $\hat{x}$. The decoder simulates the I2V scenario when receiving the condition features, while performing plain reconstruction when not. As shown in Tab. 4, we find that after the omni-objective training, our VAE can perform both plain reconstruction and image-conditioned reconstruction effectively with a single set of weights. Interestingly, because of the auxiliary information, not only does the I2V setting get enhanced quality, but we also find that this training method improves plain reconstruction. This finding shows that it is a free lunch to train Omni-AEs serving both tasks: plain reconstruction and I2V setting. Injecting these features acts as a form of conditioning augmentation: it anchors early decoder features, reduces variance in feature alignment, and regularizes the reconstruction path without introducing contradictory gradients. Because the objective remains identical for both branches, there is no source of task conflict as in typical multi-task learning. Instead, the decoder becomes more robust to variations in latent codes, improving the plain T2V setting while further benefiting I2V. For the main experimental results in the following section, unless otherwise stated, we train H3AEs with this omni-objective and inference with the plain reconstruction setting to fairly compare with baselines.

## 3.3 LATENT CONSISTENCY LOSS

Current Variational Autoencoder (VAE) training commonly employs $\mathcal{L}_1$-norm as reconstruction loss ($\mathcal{L}_{\text{recon}}$), KL as regularization ($\mathcal{L}_{\text{KL}}$) on latents, and multiple auxiliary losses ($\mathcal{L}_{\text{aux}}$) to improve reconstruction performance. This results in the following total loss:

$$\mathcal{L}_{\text{total}} = \mathcal{L}_{\text{recon}} + \lambda \mathcal{L}_{\text{KL}} + \mathcal{L}_{\text{aux}} \quad (2)$$

Existing works have investigated various auxiliary losses such as Perceptual loss (Johnson et al., 2016), Discrete Wavelet Transform (DWT) loss (Lin et al., 2024; HaCohen et al., 2024), GAN loss (HaCohen et al., 2024; Kong et al., 2024; Yang et al., 2024), *etc*. Perceptual loss measures the reconstruction error in feature space of a pre-trained neural network. Similarly, DWT loss computes the feature difference in a wavelet frequency space, while GAN loss introduces a discriminator to learn the separation between real and fake samples. In our empirical study on large-scale video VAE training, we find that the performance gains from these auxiliary losses are limited, as in Tab. 5. Plus, it is very likely to bring grid artifacts as in Fig. 3, and also noticed by HaCohen et al. (2024).

To overcome the drawbacks of these auxiliary losses, we design a new auxiliary training loss to enhance reconstruction quality by utilizing the unique paradigm of VAEs. Specifically, we reuse the VAE encoder as the discriminator, and encode the reconstructed video to obtain a fake posterior ($z'$). Note that the VAE encoder weights are not updated at this step. We compare the KL Divergence between the fake posterior ($z'$) and the posterior ($z$) encoded from the ground truth video, which has already been computed in the training forward pass. Therefore, the latent consistency loss (LC loss) is:

$$\mathcal{L}_{\text{LC}} = D_{KL}(z, z') = \frac{1}{2} \sum \left( \frac{(\mu_z - \mu_{z'})^2}{\sigma_{z'}^2} + \frac{\sigma_z^2}{\sigma_{z'}^2} - 1 - \log \frac{\sigma_z^2}{\sigma_{z'}^2} \right) \quad (3)$$

In our VAE, the latent posterior is explicitly parameterized as a Gaussian distribution with mean and variance predicted by the encoder, and sampling ($posterior.sample()$) is always performed to ensure Gaussian by construction, regardless of the KL weight used during training.

This design holds several advantages. First, we eliminate the need to incorporate an extra discriminative network as in GAN and LPIPS training. Second, empirically we found that the training is more stable without the risk of mode collapse or bringing in certain artifact patterns, as shown in the qualitative visualizations. Lastly, latent consistency loss provides a stronger guidance as the discriminator is inherited on the fly from the VAE encoder which is more powerful than a frozen network (*e.g.*, VGG for LPIPS loss). In Tab. 5, we show that our proposed latent consistency loss improves all reconstruction and fidelity metrics.

Table 3: **Comparison with SOTA Autoencoders.** Comparing with state-of-the-art autoencoders on latency, reconstruction quality on DAVIS and OpenVid datasets. For a fair comparison, we report our results without image condition (plain VAE). The I2V performance of our VAE is shown in Tab. 4 and Fig. 5. Note that our VAE is fully causal.

| Model | $f$ | #Params. (M) | FPS@$512 \times 512$ ↑ | | DAVIS | | | OpenVid-HD | | |
|---|---|---|---|---|---|---|---|---|---|---|
| | | | GPU | iPhone | PSNR↑ | SSIM↑ | rFVD↓ | PSNR↑ | SSIM↑ | rFVD↓ |
| CogVideoX-VAE | $4 \times 8 \times 8$ | 205.60 | 13.6 | ✗ | 32.37 | 0.8954 | 26.30 | 37.88 | 0.9713 | 0.81 |
| Hunyuan-VAE | $4 \times 8 \times 8$ | 235.06 | 15.5 | ✗ | 32.91 | 0.8951 | 21.17 | 39.07 | 0.9738 | 0.51 |
| Wan2.1-VAE | $4 \times 8 \times 8$ | 121.01 | 12.0 | ✗ | 32.15 | 0.8856 | 23.20 | 37.71 | 0.9674 | 0.88 |
| CV-VAE | $4 \times 8 \times 8$ | 173.99 | 24.9 | ✗ | 27.13 | 0.7722 | 220.61 | 31.78 | 0.9095 | 6.04 |
| OpenSora-1.2-VAE | $4 \times 8 \times 8$ | 375.12 | 21.7 | 14.2 | 28.76 | 0.8091 | 193.42 | 32.98 | 0.9192 | 6.93 |
| Cosmos-CV | $4 \times 8 \times 8$ | 70.74 | 141.7 | 40.5 | 31.20 | 0.8654 | 34.58 | 35.16 | 0.9492 | 1.95 |
| Cosmos-CV | $8 \times 8 \times 8$ | 107.26 | 130.8 | 42.8 | 29.34 | 0.8233 | 89.27 | 33.84 | 0.9371 | 2.51 |
| Cosmos-CV | $8 \times 16 \times 16$ | 107.26 | 161.9 | 62.0 | 26.68 | 0.7558 | 319.21 | 30.74 | 0.8916 | 15.56 |
| LTX-VAE | $8 \times 32 \times 32$ | 399.77 | 188.0 | 17.9 | 28.34 | 0.7923 | 165.43 | 34.62 | 0.9371 | 3.82 |
| H3AE | $4 \times 16 \times 16$ | 67.16 | 269.8 | 78.3 | 32.96 | 0.8987 | 20.36 | 37.43 | 0.9666 | 0.99 |
| H3AE | $8 \times 32 \times 32$ | 196.51 | 195.4 | 38.1 | 30.23 | 0.8412 | 122.82 | 35.23 | 0.9523 | 3.55 |
| H3AE | $8 \times 64 \times 64$ | 643.76 | 182.8 | 39.0 | 26.03 | 0.7404 | 688.40 | 30.23 | 0.8788 | 39.3 |

# 4 EXPERIMENTS

**Implementation details**. H3AE is trained on our internally collected image and video dataset, which has similar context statistics and aesthetics to public large-scale datasets such as Chen et al. (2024). Due to the absence of commonly adopted large-scale video dataset and different policies, most SOTA Video VAEs (Wan et al., 2025; HaCohen et al., 2024; Agarwal et al., 2025) are trained on their private dataset, making fully fair comparison difficult. To overcome this, we retrain LTX-VAE on our dataset with our training recipe, and obtain very close results, as in Appendix Tab. 8. We argue that due to the reconstruction nature, VAE training is less sensitive to video data quality and distribution. On the other hand, all design ablations in this work, including the architecture, omni-objective training and latent consistency loss are trained on the same data and recipe so fair comparison is ensured. Our model is trained on $256 \times 256$ video clips with various length from 1 to 49 for 80K iterations. Latent consistency loss is only applied in the final 10K iters. The training is conducted on 32 NVIDIA A100 80G GPUs. We use AdamW optimizer with $1e-4$ learning rate and $\beta = [0.9, 0.999]$, and reduce the learning rate to $1e-5$ in the last 10K iters.

**Evaluation**. We benchmark reconstruction quality using high resolution video dataset, including 60 video clips from DAVIS-2017 (Perazzi et al., 2016) *testset* and 4000 high resolution video clips randomly sampled from OpenVid (Nan et al., 2025) Our training data does not overlap with DAVIS or OpenVid, so that zero-shot evaluation is strictly enforced. The first 33 frames from each clip are utilized for evaluation with a spatial resolution at $512 \times 512$. We evaluate the PSNR, SSIM, and reconstruction-FVD (rFVD) to benchmark the performance of autoencoders.

## 4.1 COMPARISON WITH SOTA AUTOENCODERS

We compare our model with SOTA video tokenizers, *i.e.* CogVideoX-VAE, CV-VAE (Zhao et al., 2024a), Cosmos-Tokenizer (Agarwal et al., 2025), LTX-VAE (HaCohen et al., 2024), etc. Among them, Cosmos-Tokenizer and LTX-VAE are focused on high compression ratios. Three variants of H3AE models are built up with $4 \times 16 \times 16$, $8 \times 32 \times 32$, and $8 \times 64 \times 64$ compression ratios respectively to demonstrate the robustness of our design choices and analysis. We obtain the pre-trained SOTA models and evaluate them under the same evaluation setting mentioned above for reconstruction quality. We report inference speed on Nvidia A100 GPU and iPhone 16 Pro Max.

As shown in Tab. 3 and Fig. 1, our models achieve better reconstruction under the same or higher compression ratio. Specifically, with $4\times$ higher compression ratio, our $4 \times 16 \times 16$ VAE outperforms Cosmos-Tokenizer ($4 \times 8 \times 8$) across both evaluation sets, achieving notable improvements of $+1.76$ PSNR, $+0.0333$ SSIM, $-14.22$ rFVD on DAVIS, as well as $+2.27$ PSNR, $+0.0174$ SSIM, $-0.96$ rFVD on OpenVid-HD. Furthermore, our $8 \times 32 \times 32$ VAE significantly outperforms LTX-VAE by $+1.89$ PSNR, $+0.048$ SSIM, $-42.61$ rFVD on DAVIS, along with $+0.61$ PSNR, $+0.0152$ SSIM, $-0.27$ rFVD on OpenVid-HD. In addition, our model demonstrates superior overall efficiency. Our $8 \times 32 \times 32$ VAE requires only half the parameters compared to LTX-VAE, and achieves $2.5\times$ speedup on iPhone.

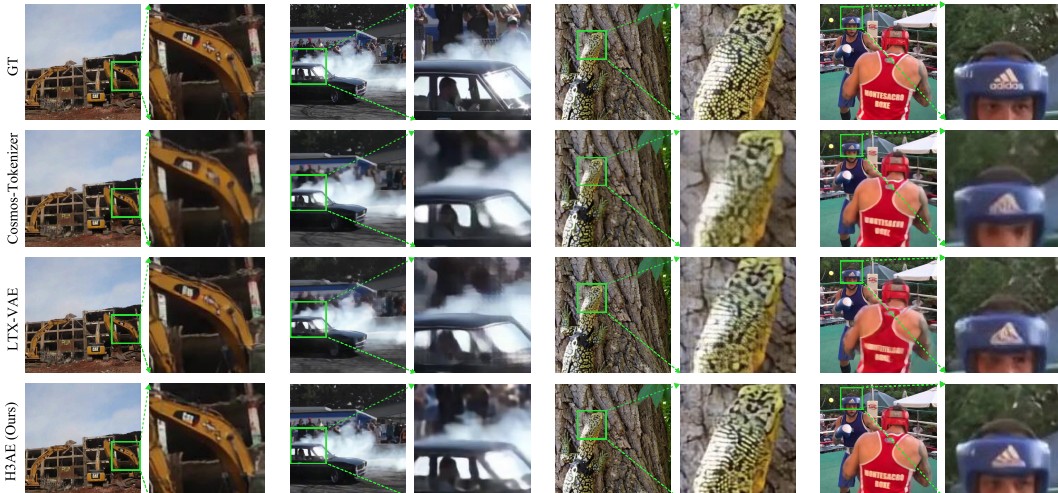

Figure 3: **AE Qualitative Results.** Reconstructions from our H3AE ($8 \times 32 \times 32$) and other high compression autoencoders: Cosmos-Tokenizer (Agarwal et al., 2025) ($8 \times 16 \times 16$), LTX-VAE (Ha-Cohen et al., 2024) ($8 \times 32 \times 32$). We show zoomed-in results to highlight the differences in fidelity and quality. Our method features greater high-frequency detail. GT refers to the ground truth video.

**Visual Quality**. We exhibit a visual comparison of reconstructed samples between Cosmos-Tokenizer ($8 \times 16 \times 16$), LTX-VAE, and our H3AE ($8 \times 32 \times 32$) in Fig. 3. The results demonstrate that our model delivers better contextual fidelity and high-frequency quality than Cosmos-Tokenizer. Noteworthy, the visual comparison also demonstrates the effectiveness of our proposed training method, while Cosmos-Tokenizer and LTX-VAE either introduces overly smooth reconstruction or unpleasant artifacts.

## 4.2 ABLATION STUDY

We evaluate the effectiveness of the proposed omni-objective training in Tab. 4. Compared to the baseline, our approach yields substantial improvements in the image-conditioned I2V setting, while also enhancing the plain T2V setting as a byproduct—effectively providing a "free" gain in reconstruction quality.

We further ablate the proposed latent consistency loss in Tab. 5. Results show that it consistently outperforms commonly used auxiliary objectives, such as perceptual and adversarial losses, on both reconstruction accuracy and fidelity metrics, while maintaining greater training stability.

Table 4: **Omni-AE for the T2V and I2V tasks**. The baseline is trained without image condition while omni-training utilized the image condition with probability ($p = 0.5$).

| Our VAE | PSNR↑ | SSIM↑ | LPIPS↓ | FloLPIPS↓ | rFVD↓ |
|---|---|---|---|---|---|
| $8 \times 32 \times 32$ | 29.48 | 0.8280 | 0.2627 | 0.2611 | 147.1 |
| omni-plain | 29.95 | 0.8367 | 0.2506 | 0.2470 | 129.6 |
| omni-I2V | 30.08 | 0.8384 | 0.2456 | 0.2427 | 128.4 |
| $8 \times 64 \times 64$ | 25.21 | 0.7284 | 0.3937 | 0.4048 | 737.9 |
| omni-plain | 25.97 | 0.7387 | 0.3963 | 0.4041 | 709.7 |
| omni-I2V | 26.38 | 0.7503 | 0.3748 | 0.3879 | 684.1 |

Table 5: **Training loss comparison**. Baseline is $L_{\text{recon}} + \lambda L_{\text{KL}}$. Auxiliary losses and our latent consistency loss are applied to the same base model and trained for 10K iters for comparison.

| Loss | PSNR↑ | SSIM↑ | LPIPS↓ | FloLPIPS↓ | rFVD↓ |
|---|---|---|---|---|---|
| Baseline | 29.48 | 0.8280 | 0.2627 | 0.2611 | 147.1 |
| LPIPS | 29.07 | 0.8205 | 0.224 | 0.2275 | 148.9 |
| GAN | 29.35 | 0.8241 | 0.2631 | 0.2623 | 161.0 |
| DWT | 29.36 | 0.8247 | 0.2651 | 0.2642 | 159.3 |
| Ours | 29.58 | 0.8299 | 0.2541 | 0.2511 | 142.2 |

## 4.3 DIFFUSION GENERATION IN H3AE LATENT SPACE

We show text-to-video generation results in Tab. 6 and Fig. 4 to demonstrate that our VAE creates a good and highly-compressed latent space for video diffusion models. We construct a 2B DiT aligned with popular video diffusion model settings. Specifically, we use 3D full attention with RoPE and QK-Norm in transformer blocks. All models are first trained on image dataset to learn

spatial knowledge for $50K$ iterations, then these model are trained with image-video joint training strategy for $100K$ iterations. The VDM training is done on 128 A100 GPUs for 4-10 days depending on the VAE compression ratio. As shown in Tab. 6, we find that using our H3AE ($8 \times 32 \times 32$) outperforms Cosmos-Tokenizer ($8 \times 16 \times 16$) and LTX-VAE in Vbench score. To isolate the impact of the training dataset, we report the scores of both the official LTX model and our finetuned version. Our high-compression ratio autoencoder also enables fast and memory-efficient inference on both GPU and mobile. We additionally report the result of using an extremely high-compression ratio ($8 \times 64 \times 64$) H3AE, which provides more than a $4 \times$ speed-up on the iPhone 16 PM, which demonstrates the great potential of high compression VAEs for efficient video generation.

Table 6: **Quantitative Metrics on VBench.** Text-to-video generation benchmark on Vbench (Huang et al., 2024) with a 2B DiT denoiser trained using different Autoencoders. Overall Vbench score and selected score are reported. We specifically exhibit aesthetic quality (**AQ**), imaging quality (**IQ**), and motion smoothness (**MS**) to evaluate the performance of denoisers. Latency of DiT is benchmarked by generating **65-frame** $512 \times 512$ video clips on an NVIDIA A100 80GB GPU and **17-frame** $512 \times 512$ clips on iPhone 16 Pro Max. Notably the DiT using Cosmos-Tokenizer results in **OOM** error on iPhone due to memory inefficiency.

| AE | $|z|$ | $f$ | DiT Time (s)↓ | | Vbench Score↑ | | | | | |
|---|---|---|---|---|---|---|---|---|---|---|
| | | | GPU | iPhone | AQ | IQ | MS | Quality | Semantic | Total |
| Cosmos-Tokenizer | 16 | $8 \times 16 \times 16$ | 0.40 | ✗ | 0.5606 | 0.5097 | 0.9875 | 0.8101 | 0.6860 | 0.7853 |
| LTX-VAE | 128 | $8 \times 32 \times 32$ | 0.09 | 1.00 | 0.5981 | 0.6028 | 0.9896 | 0.8230 | 0.7079 | 0.8000 |
| LTX-VAE[1] | 128 | $8 \times 32 \times 32$ | 0.09 | 1.00 | 0.6151 | 0.6254 | 0.9921 | 0.8208 | 0.7209 | 0.8008 |
| H3AE | 128 | $8 \times 32 \times 32$ | 0.09 | 1.00 | 0.6017 | 0.6073 | 0.9875 | 0.8342 | 0.7147 | 0.8103 |
| H3AE | 256 | $8 \times 32 \times 32$ | 0.09 | 1.00 | 0.6170 | 0.6314 | 0.9885 | 0.8338 | 0.7226 | 0.8110 |
| H3AE | 256 | $8 \times 64 \times 64$ | 0.05 | 0.22 | 0.5441 | 0.5317 | 0.9901 | 0.8003 | 0.6153 | 0.7633 |

[1]LTX-Video DiT finetuned on our datasets.

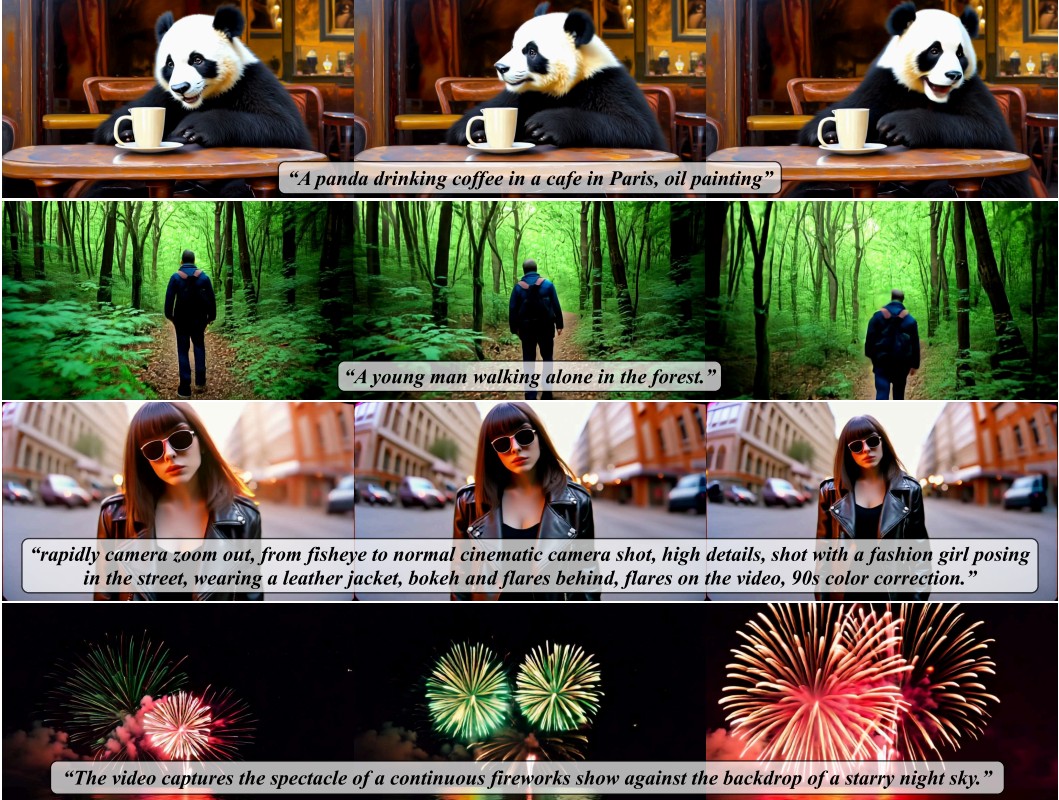

*"A panda drinking coffee in a cafe in Paris, oil painting"*

*"A young man walking alone in the forest."*

*"rapidly camera zoom out, from fisheye to normal cinematic camera shot, high details, shot with a fashion girl posing in the street, wearing a leather jacket, bokeh and flares behind, flares on the video, 90s color correction."*

*"The video captures the spectacle of a continuous fireworks show against the backdrop of a starry night sky."*

Figure 4: **T2V Qualitative Results.** Examples of videos generated by a 2B DiT denoiser, trained on the latent space of our $8 \times 32 \times 32$ H3AE.

### 4.4 H3AE AS IMAGE AUTOENCODER

The H3AE can be used as an image VAE and its image reconstruction performance is presented in Tab. 7. Although H3AE targets a video autoencoder, it can still be utilized as an image autoencoder. We compare H3AE's performance with current SOTA high-compression autoencoder DC-AE (Chen et al., 2025b), showing that H3AE outperforms DC-AE under the same compression ratios Tab. 7.

Table 7: Despite H3AE is designed as video autoencoder, it can also be used as an image autoencoder. The quality comparison of image reconstruction between current SOTA high-compression autoencoder *i.e.* Chen et al. (2025b) and our H3AE on DAVIS and OpenVid-HD datasets. The results demonstrate H3AE outperforms DC-AE under same compression ratios.

| AE | $f$ | Params | DAVIS | | | OpenVid-HD | | |
|---|---|---|---|---|---|---|---|---|
| | | | PSNR | SSIM | LPIPS | PSNR | SSIM | LPIPS |
| DC-AE | $32 \times 32$ | 308.4 | 28.28 | 0.7886 | 0.0732 | 30.31 | 0.8709 | 0.0474 |
| H3AE | $32 \times 32$ | 196.5 | 36.27 | 0.9475 | 0.0724 | 38.11 | 0.9694 | 0.0378 |
| DC-AE | $64 \times 64$ | 645.5 | 28.14 | 0.7865 | 0.0746 | 30.16 | 0.8689 | 0.0479 |
| H3AE | $64 \times 64$ | 643.8 | 29.50 | 0.8169 | 0.2640 | 31.04 | 0.8822 | 0.1695 |

### 4.5 H3AE AS IMAGE-TO-VIDEO AUTOENCODER

Figure 5 demonstrates the performance of our omni-objective training in the image-to-video (I2V) setting. By randomly dropping the first-frame condition during training, H3AE learns to flexibly decode with or without image guidance. As shown, conditioning on the first frame enables the VAE to better preserve details, while still maintaining robustness when the condition is absent. This demonstrates that omni training not only unifies T2V and I2V within a single model but also enhances reconstruction quality in both regimes.

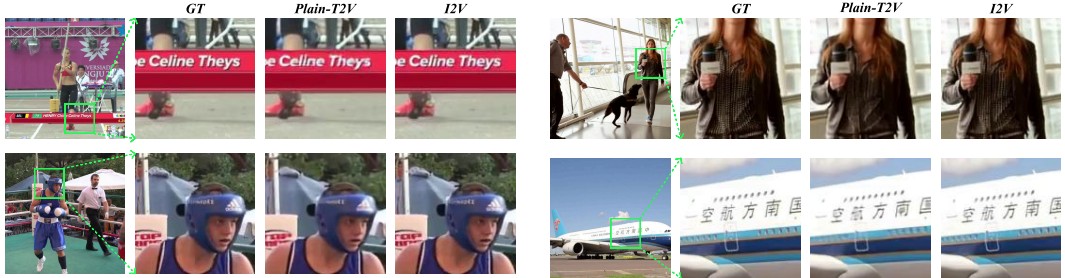

Figure 5: Quality comparison of reconstruction results of H3AE between plain-T2V VAE and I2V VAE settings. The results shows that I2V VAE delivers better high-frequency details.

## 5 CONCLUSION

In this work, we systematically examine autoencoder architecture design and optimize the computation distribution to obtain a series of efficient AEs that can decode latents in real-time on mobile device. With the ultra high spatial-temporal compression ratio, we successfully reduce the latent tokens and achieve faster generation speed for the DiT-based video diffusion model. We unify plain reconstruction and image-conditioned I2V reconstruction, demonstrating improved results for both settings with a single set of weights, simplifying applications and saving the potential adaptation cost. Our empirical study also shows that popular discriminative losses, i.e., GAN, LPIPS, and DWT losses, provide no significant improvement when training AEs at scale. We propose a novel latent consistency loss that does not require complicated discriminator design or hyperparameter tuning but provides stable improvements in reconstruction quality. We discuss limitations and broader impact in Appendix B.

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

## A  USE OF LLMs

Large language models (e.g., ChatGPT, Gemini) were used exclusively for grammar polishing and formatting assistance. All proposed concepts, experiment design, and analysis are NOT generated by LLMs.

## B  LIMITATIONS AND BROADER IMPACT

While H3AE advances the design and training of video VAEs, several limitations remain. Though we demonstrate mobile deployment, our evaluations are still bounded by current hardware and model scales; larger-scale models may reveal new bottlenecks. Additionally, both VAE and DiT training assume access to large-scale video datasets, which are not universally available. We hope that this can be standardized for the research community in the near future.

For the broader impact of H3AE, efficient VAE makes high-quality video generation more accessible by reducing the computational and memory demands of diffusion models. This democratizes research and creative applications, enabling use on consumer devices and in resource-constrained environments. However, the same accessibility raises concerns about misuse, such as the large-scale generation of misleading or harmful content.

## C  EXPERIMENT FAIRNESS.

To eliminate the impact of different data distributions and ensure fair comparisons, we re-trained the LTX VAE (HaCohen et al., 2024) on the same dataset and under the same training setup as H3AE, as in Tab. 8. We find that with our video dataset and training recipe, similar results are obtained compared to the official LTX VAE weights. Due to different training stages and loss weights, our retrained LTX VAE is slightly better at reconstruction metrics but a bit worse in rFVD. Due to the reconstruction nature, Video VAE training does not rely a lot on video data distribution and quality. Our dataset and training strength is on par with public Video VAE works and thus produce comparable results.

Table 8: Dataset ablation.

| Model | $f$ | PSNR↑ | SSIM↑ | rFVD↓ |
|---|---|---|---|---|
| LTX-VAE | $8 \times 32 \times 32$ | 28.34 | 0.7923 | 165.43 |
| LTX-VAE-Ours | $8 \times 32 \times 32$ | 28.78 | 0.8133 | 185.33 |
| H3AE | $8 \times 32 \times 32$ | 30.23 | 0.8412 | 122.82 |

## D  MODEL ARCHITECTURE.

We provide the details of the optimized H3AE architecture in Tab. 9. Note that the spatial stage only handles spatial downsample and upsample, and the spatial-temporal stage applies either or both spatial and temporal sampling according to the designated configuration. For instance, for the $8 \times 32 \times 32$ setting, there are two $1 \times 2 \times 2$ downsamplings in the spatial stage and three $2 \times 2 \times 2$ downsamplings in the spatial-temporal stage.

In this work, we use 3D convolutions with kernel size 3 and repeated causal padding. Downsampling is done via strided convolution, while upsampling is ablated in the main paper.

Table 9: H3AE architecture reported with channel dimension and number of blocks. Note that there are two convolution layers in a residual block. S and ST refer to spatial and spatial-temporal stages, respectively. We use 8 heads for the causal attention.

| Compression Ratio | S Stage-1 Conv2D | S Stage-2 Conv2D | ST Stage-1 Conv3D | ST Stage-2 Conv3D | ST Stage-3 Conv3D | ST Stage Attn. |
|---|---|---|---|---|---|---|
| $4 \times 16 \times 16$ | $32 \times 2$ | $64 \times 2$ | $128 \times 2$ | $256 \times 8$ | - | - |
| $8 \times 32 \times 32$ | $32 \times 2$ | $64 \times 2$ | $128 \times 2$ | $256 \times 6$ | - | $512 \times 8$ |
| $8 \times 64 \times 64$ | $32 \times 2$ | $64 \times 2$ | $128 \times 2$ | $256 \times 6$ | $512 \times 8$ | $512 \times 8$ |

## E   DESIGN OF LATENT CONSISTENCY LOSS

We employ KL divergence for Latent Consistency Loss because it directly reflects the true probabilistic structure of a variational autoencoder, where the encoder outputs a Gaussian posterior $q(z|x) = \mathcal{N}(\mu, \sigma^2)$ and the latent is explicitly sampled according to the mean $\mu$ and variance $\sigma^2$ of the posterior in a non-deterministic manner.

Alternatively, we can use the entire VAE to reconstruct the fake sample instead of using fake posterior, in which case L1/L2 loss is applicable, and the method becomes "pixel consistency loss". Note that this setting further slows down training because of the additional decoder forward, while in our experiments, we find it offers no further gain compared to the encoder-KL setting, as in Tab. 10. As a result, we stick to the latent consistency setting in this work.

| Loss | PSNR↑ | SSIM↑ | rFVD↓ |
|------|-------|-------|-------|
| Baseline | 29.48 | 0.8280 | 147.1 |
| Latent Consistency | 29.58 | 0.8299 | 142.2 |
| Pixel Consistency | 29.52 | 0.8286 | 143.9 |

Table 10: Design of consistency loss.

## F   ANALYSIS OF TEMPORAL SMOOTHNESS

In our experiments, we do not observe significant temporal flickering artifacts in either our method or the baseline video VAEs, as video VAEs generally relies on 3D convolutions to ensure local coherence. Instead, quality degradation under extreme compression is mainly attributed to per-frame high-frequency details and mild blurriness, particularly in fast-motion scenarios as in the example in Fig. 6. The visual results are consistent with the quantitative metrics in Tab. 3 and Fig. 1, where H3AE shows better temporal stability and reconstruction fidelity.

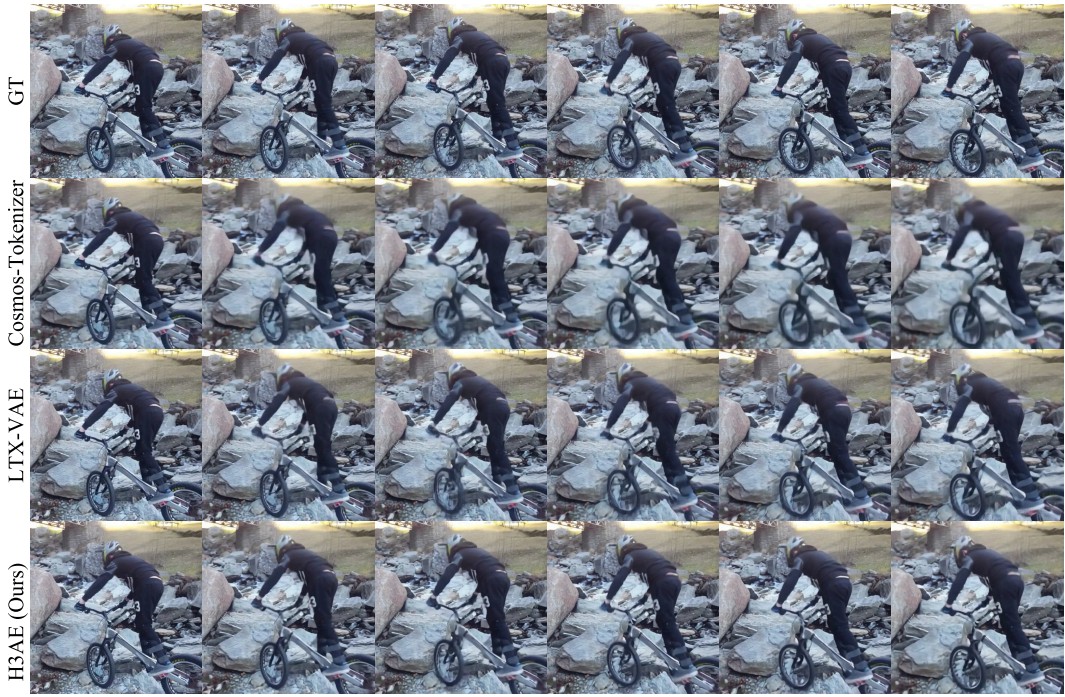

Figure 6: Visualization of temporal smoothness.

## G  MORE VIDEO VISUALIZATIONS

We provide more video visualizations of the diffusion transformer trained under H3AE latent space in Fig. 7 and the *supplementary files*.

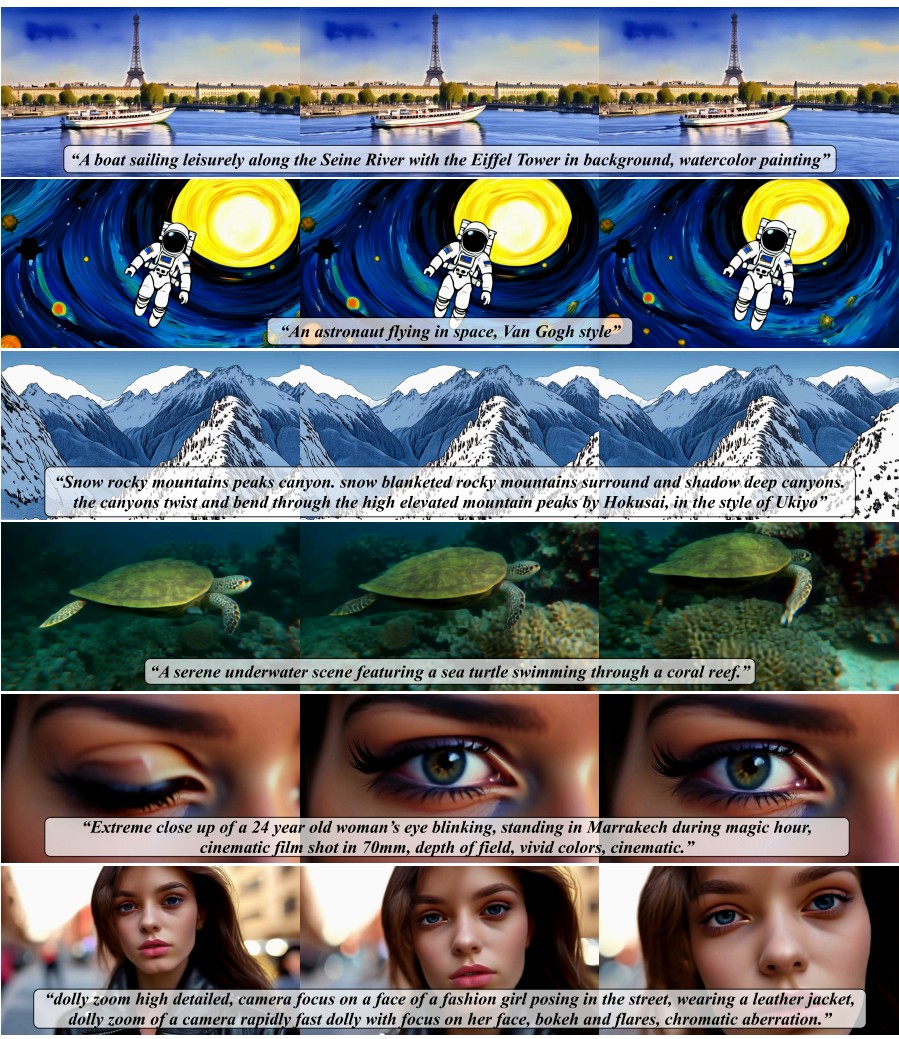

Figure 7: More videos generated by a 2B DiT denoiser, trained on the latent space of our H3AE.

