# OpenReview forum: "H3AE: High Compression, High Speed, and High Quality AutoEncoder for Video Diffusion Models"
_ICLR.cc/2026/Conference — Submitted to ICLR 2026_

### Official Review · Reviewer_DffG · 2025-10-31

**Soundness:** 2
**Presentation:** 2
**Contribution:** 2
**Rating:** 2
**Confidence:** 4

**Summary:**

This paper examines the design of autoencoder architectures and proposes an Omni-AE training strategy to improve efficient video generation. A latent consistency loss function is designed to enhance the perceptual and pixel fidelity of the reconstructed results. The proposed method achieves ultra-high compression ratios and real-time decoding speeds on GPUs and mobile devices, surpassing other competing methods by a large margin.

**Strengths:**

The proposed method aims to enable real-time I2V and T2V on GPUs and mobile devices. The manuscript is well-organized.

**Weaknesses:**

[1].H3AE should be explained in line 24 of the manuscript because it appears for the first time in the paper.
[2].The paper lacks the details of H3AE architecture. For example, the kernel size used in the convolution. How to downsamle the features?
[3].In line 184, what is RoPE and QK-norm? Please explain this and cite the related papers.
[4].In line 191, what is 4ch? This content is unreadable.
[5].In Eq.(1), how to set the value of p during the training phase?
[6].How to use the hierarchical features of the first frame? Please explain this. How about using cross-attention or other ways can improve the performance?
[7].In Fig. 3, the visual comparison in the last column is not obvious. Please change this.

**Questions:**

See the part of weaknesses.

---

> ### Author Response · Authors · 2025-11-26
> **Author Response**
>
> **1. H3AE should be explained in line 24 of the manuscript because it appears for the first time in the paper.**
>
> Thanks for pointing out, since it is in the title, we underline it as highlighted in the revision.
>
> **2. The paper lacks the details of H3AE architecture. For example, the kernel size used in the convolution. How to downsample the features?**
>
> In Appendix D, we provided the architecture specification. We use kernel size 3 for all convolution layers, and the feature is downsampled through strided convolutions. (Upsampling design is ablated as in the main paper.) We add these details as highlighted in the revision.
>
> **3. In line 184, what is RoPE and QK-norm? Please explain this and cite the related papers.**
>
> Thanks for the comment. We have added the corresponding citations and explanations as highlighted in line#183 in the revision.
>
> **4. In line 191, what is 4ch?**
>
> Thanks for the suggestion, we revert the abbreviation and use 4-channel in the revision (aligned with line #198-199).
>
> **5. In Eq.(1), how to set the value of p during the training phase?**
>
> In our experiments we set p as 0.5. We include an ablation study on p values as below. We find that setting the probability to different values gives quite similar results and the performance gain from Omni-objective training is stable. We attribute this to the training scale of video. With sufficient training iterations, the model gets adapted to both conditions, i.e., with or without the leakage of additional image information.
>
> |  8x32x32 |  PSNR |  SSIM  |  LPIPS | FloLPIPS |  rFVD |
> |:--------:|:-----:|:------:|:------:|:--------:|:-----:|
> | Baseline | 29.48 | 0.8280 | 0.2627 |  0.2611  | 147.1 |
> |  p=0.25  | 29.95 | 0.8365 | 0.2499 |  0.2451  | 130.1 |
> |   p=0.5  | 29.95 | 0.8367 | 0.2506 |  0.2470  | 129.6 |
> |  p=0.75  | 29.93 | 0.8365 | 0.2522 |  0.2481  | 129.1 |
>
> **6. How to use the hierarchical features of the first frame? Please explain this. How about using cross-attention or other ways can improve the performance?**
>
> As in Eq.1, we add it to the features of the corresponding decoder stage with identical weight (i.e., taking average). Cross attention does not become our choice because in terms of computation and peak memory, it is not affordable at higher resolution stages because of the enormous number of tokens, which does not meet our high-speed mobile-ready scenario.
>
> **7. In Fig. 3, the visual comparison in the last column is not obvious. Please change this.**
>
> Thanks for the suggestion, we update it with another sample as in the revision.

---

> > ### Comment · Reviewer_DffG · 2025-11-28
> >
> > Thank you for your response. There is one remaining question:
> > Could you please explain the RoPE and the QK-norm in detail?

---

> > > ### Author Response · Authors · 2025-12-04
> > >
> > > Thanks for your comment. We have included the reference work and descriptions as highlighted in line#183 - line#189 in the revision.
> > >
> > > We adopt Rotary Positional Embeddings (RoPE) to encode positional information: instead of adding fixed absolute positional vectors, we rotate the query and key embeddings by an angle determined by the token’s position. This geometry-based encoding causes the dot product between any two positions to depend only on their relative distance, naturally capturing ordering and enabling better generalization to different resolutions, especially when sequence lengths vary or exceed those used during training.
> > >
> > > We apply Query-Key Normalization (QK-Norm) to stabilize training and accelerate convergence. Before computing attention logits we normalize the query and key vectors so that the magnitude of the attention scores are constrained and raw amplitude is avoid. QK-Norm and RoPE are widely employed in recent SOTA Transformers (e.g., CogVideoX, Wan, Hunyuan, LTX, etc.).

---

### Official Review · Reviewer_WhDk · 2025-10-31

**Soundness:** 3
**Presentation:** 3
**Contribution:** 3
**Rating:** 4
**Confidence:** 5

**Summary:**

This paper proposes a series of video auto-encoders with high spatial compression and with real-time mobile decoding capability.
It combines architectural engineering choices such as 2D spatial convs at highest resolution and 3D causal attention. It also proposes an "omni-objective" training scheme that allows the VAE to be used as plain as well as for I2V tasks. Finally it also proposes a latent-consistency loss to improve quality.  Along with quantitative and qualitative results for VAE, authors also train a text-to-video DiT model to showcase the latent space capability of the trained VAEs.

**Strengths:**

This paper is a very well executed engineering effort that proposes a new Video VAE architecture and improvement to the training pipeline. This is specifically also tailored towards real-time decoding on mobile devices.

1. The paper proposes a modified architecture design for practical VAEs.  The VAE's have significantly improved performance, particularly the one with compression ratio of $4 \times 16 \times 16$.

2. The idea of training a single VAE that can be used both as a plain VAE and for I2V scenario is pretty neat.

3. The architecture ablation study is well-detailed and the final architecture proposed seems to be quite efficient on hardware with a high FPS (Table-1).

**Weaknesses:**

1. The idea of conditioning decoder on image information has been explored before. Authors propose to train a VAE that can be used for both T2V and I2V tasks, However, there are no experiments that verify the usefulness of the VAE for the I2V denoiser scenario.

2. The quality of VAE latents is mainly measured with quantitative and qualitative samples. However, since this is a Video VAE, there is not much discussion / analysis about the temporal consistency and any flickering artifacts that may arise from such extreme compression.

3. A user study that compares the various compression ratios is also missing.

4. Comparision of SoTA models in Table-1 seems unfair since the compression ratios are quite different. It makes it difficult to really judge if techniques proposed in this paper really are much more beneficial than the other methods.

**Questions:**

1. Authors have provided ablation studies for each improvement in Tab. 2, 4, 5 . However, could the authors provide a clear explanation of which component (architecture improvements vs latent consistency vs omni-AE training ) provided the most gains in terms of performance (PSNR)? Is the performance of each component similar across different compression ratios?

2. SoTA comparisions with efficient decoders such as [Turbo-VAED Zou et. al](https://arxiv.org/pdf/2508.09136) are missing.

3. The latent consistency loss is posed as a novel contribution. However, it bears a close resemblance to regularization applied in the paper [Consistency Regularisation for Variational Auto-encoders Sinha et. al](https://arxiv.org/pdf/2105.14859). Could the authors provide more clarity to this? It seems like a common regularisation technique being used in classifiers.

4. Did the authors also attempt high temporal compression along with spatial compression? Did they observe any significant performance changes?

---

> ### Author Response · Authors · 2025-11-26
> **Author Response 1/2**
>
> **1. The idea of conditioning decoder on image information has been explored before. Authors propose to train a VAE that can be used for both T2V and I2V tasks, However, there are no experiments that verify the usefulness of the VAE for the I2V denoiser scenario.**
>
> 1. Thanks for the question. We first clarify that the purpose of the proposed omni-objective training is to enable a single VAE to support both T2V and I2V reconstruction paths. This is fundamentally different from prior dedicated I2V VAEs (e.g., Reducio), which are not compatible with the plain T2V setting and require separate models.
> 2. Second, regarding the usefulness of the VAE for I2V, Table 4 and Fig. 5 already include quantitative and qualitative I2V reconstruction results that directly evaluate this capability. These results demonstrate that omni-objective training consistently improves the I2V reconstruction quality over a plain VAE, confirming that H3AE can effectively leverage first-frame conditioning. Training a full I2V diffusion model: (i) is beyond the scope of this work (ii) does not provide additional validation of I2V VAE. Given an I2V denoiser, we still feed the generated latent into H3AE with or without the additional image condition, and the relative improvements remain identical to those shown in Table 4 and Fig. 5. Therefore, we evaluate the I2V case in the correct context of autoencoding, independent from the I2V diffusion generator.
>
> **2. The quality of VAE latents is mainly measured with quantitative and qualitative samples. However, since this is a Video VAE, there is not much discussion / analysis about the temporal consistency and any flickering artifacts that may arise from such extreme compression.**
>
> We appreciate the comment. In our experiments, we do not observe significant temporal flickering artifacts in either our method or the baseline VAE. As video VAEs relies on 3D convolutions to ensure local coherence. Instead, quality degradation under extreme compression is mainly attributed to per-frame high-frequency details and mild blurriness, particularly in fast-motion scenarios as in the example in Appendix F. We have added visual comparisons in the supplementary that clearly illustrate this behavior. The visual results are consistent with the quantitative metrics in Table 3, where H3AE shows better temporal stability and reconstruction fidelity.
>
> **3. A user study that compares the various compression ratios is also missing.**
>
> We would like to clarify that user studies are typically essential for evaluating the perceptual quality of generative models, but are not the standard form of evaluating VAE reconstruction (e.g., LTX, Cosmos, CV-VAE, etc.). VAE performance is dominated by pixel-level similarity to ground truth, which are better captured through quantitative metrics such as PSNR, SSIM, and rFVD. As shown in Table 3, these metrics already provide a clear and consistent quality–compression ratio trade-off (e.g., 4×16×16 vs. 8×32×32). A user study would largely mirror these quantitative trends.
>
> **4. Comparision of SoTA models in Table-1 seems unfair since the compression ratios are quite different. It makes it difficult to really judge if techniques proposed in this paper really are much more beneficial than the other methods.**
>
> We thank the reviewer for the comment, but we believe the comparison is fair and clear once viewed through the correct lens: we push forward the Pareto frontier of compression-quality tradeoff, as visualized in Fig.1. At the same compression ratio, H3AE achieves higher reconstruction quality than prior methods; and at the same quality level, H3AE attains substantially higher compression, which highlights the superior performance of H3AE.

---

> ### Author Response · Authors · 2025-11-26
> **Author Response 2/2**
>
> **5. Authors have provided ablation studies for each improvement in Tab. 2, 4, 5 . However, could the authors provide a clear explanation of which component (architecture improvements vs latent consistency vs omni-AE training ) provided the most gains in terms of performance (PSNR)? Is the performance of each component similar across different compression ratios?**
>
> Thanks for the question. We summarize the component-wise improvements using PSNR and rFVD across tables. The architectural design contributes the dominant gain (Table 1), improving PSNR from (24~28)-baselines to 29.48 dB by enabling more effective spatio-temporal representation under high compression. On top of this, omni-objective training brings a further improvement of +0.47 dB (Table 4), and the proposed latent consistency loss adds +0.10 dB PSNR and –4.9 rFVD (Table 5), enhancing fine-detail sharpness while preserving temporal stability. The gains from both omni-objective training and LC loss are consistent across the evaluated compression ratios (8×32×32 and 8×64×64), as shown in Tables 4 and 5. We add ablations of latent consistency loss across different compression ratios to extend Table 5, as follows.
>
> | Compression |   Loss   |  PSNR |  SSIM  |  LPIPS | FloLPIPS |  rFVD |
> |:-----------:|:--------:|:-----:|:------:|:------:|:--------:|:-----:|
> |   4x16x16   | Baseline | 32.34 | 0.8899 | 0.1583 |  0.1538  |  40.3 |
> |   4x16x16   |   Ours   | 32.46 | 0.8896 | 0.1583 |  0.1527  |  28.8 |
> |   8x32x32   | Baseline | 29.48 | 0.8280 | 0.2627 |  0.2611  | 147.1 |
> |   8x32x32   |   Ours   | 29.58 | 0.8299 | 0.2541 |  0.2511  | 142.2 |
> |   8x64x64   | Baseline | 25.21 | 0.7284 | 0.3937 |  0.4048  | 737.9 |
> |   8x64x64   |   Ours   | 25.33 | 0.7301 | 0.3742 |  0.3890  | 639.0 |
>
> **6. SoTA comparisons with efficient decoders such as Turbo-VAED Zou et. al are missing.**
>
> Thanks for the suggestion, we have added the citation to Turbo-VAED as highlighted in the revision. Turbo-VAED is a very recent arXiv preprint that appeared concurrently with this submission and was not available during the design and execution of our experiments. By inspecting the paper, we find that their proposed (i) usage of depth-wise 3D convolution, (ii) 3D pixel shuffle, and (iii) decoder distillation are conceptually orthogonal to our contributions, which focus on VAE block design and distribution, omni-objective training, and the latent consistency loss. These approaches are complementary rather than compare-and-compete.
>
> **7. The latent consistency loss is posed as a novel contribution. However, it bears a close resemblance to regularization applied in the paper Consistency Regularisation for Variational Auto-encoders Sinha et. al. Could the authors provide more clarity to this? It seems like a common regularisation technique being used in classifiers.**
>
> Our latent consistency loss is fundamentally different from prior consistency-regularized VAEs (e.g., Sinha & Dieng 2021). Prior work enforces augmentation-based invariance by minimizing KL divergence between the posterior of an image and that of its semantic-preserving transformation (e.g., rotation/translation). In contrast, our method compares the posterior of the ground-truth video with the posterior of the model reconstruction (“fake posterior”), obtained by re-encoding the reconstructed video with a frozen encoder. Thus, our loss acts as a self-distillation signal that improves reconstruction fidelity, not an invariance constraint. Moreover, unlike prior consistency methods that operate on static images and target representation robustness, our loss is designed specifically for high-compression video VAEs to outperform artifact-prone auxiliary losses (LPIPS/GAN/DWT) and stabilize training. Therefore, while both works involve a KL term in latent space, the objective and mechanism are fundamentally different.
>
> **8. Did the authors also attempt high temporal compression along with spatial compression? Did they observe any significant performance changes?**
>
> We thank the reviewer for the question. We address this from two angles:
> 1. Our models already explore higher temporal compression in addition to spatial compression: H3AE variants are trained at 4×16×16, 8×32×32, and 8×64×64 compression (temporal factor 4 or 8), as reported in Table 3 and Fig. 1. As the overall compression increases (including the temporal dimension), we see the expected trade-off: PSNR/SSIM and rFVD degrade moderately, but H3AE still remains on a more favorable compression–quality Pareto frontier than prior VAEs.
> 2. Additionally, we include a 8x16x16 VAE to align the spatial compression with the reported 4x16x16 one but only compare temporal compression.
>
> |      Model     |  PSNR |  SSIM  |  rFVD  |
> |:--------------:|:-----:|:------:|:------:|
> |     4x16x16    | 32.96 | 0.8987 |  20.36 |
> |     8x16x16    | 32.31 | 0.8842 |  43.37 |
> | Cosmos-8x16x16 | 26.68 | 0.7558 | 319.21 |

---

### Official Review · Reviewer_rERM · 2025-11-06

**Soundness:** 2
**Presentation:** 2
**Contribution:** 2
**Rating:** 2
**Confidence:** 4

**Summary:**

This paper presents H3AE, an efficient and high-compression video autoencoder. The authors systematically explore VAE architectural components and introduce a LC Loss and Omni-objective training strategy, achieving strong improvements in reconstruction quality.

**Strengths:**

1. The paper provides a systematic study of video VAE design choices e.g., attention and normalization.
2, H3AE achieves notable improvements in recosntruction over strong baselines.

**Weaknesses:**

1. This paper is an engineering work. The main contributions stem from combining existing components, e.g., 2D spatial convolutions in high-resolution stages, 3D causal attention, and increased latent channels—without introducing new conceptual insights.

2. The paper does not provide explanations for key findings, remaining at an empirical reporting level. For instance, it does not clarify why the proposed omni-objective training yields “free lunch” improvements instead of task conflict between t2v and I2V objectives.

3. While reconstruction metrics improve, generation results (e.g., Fig. 6) show little gain over baselines, suggesting limited impact on generative capability. Reconstruction and generation quality are not highly correlated; for generative tasks, improvements in generation metrics would be more meaningful.

4. It remains unexplained why the latent consistency loss is formulated as a KL divergence rather than a simpler distance such as MSE. Additional ablation studies are needed to justify this choice.

5. The code is not released, which reduces transparency and reproducibility for a work that is primarily engineering-oriented.

**Questions:**

See Weaknesses. Overall, I believe this paper presents valuable engineering exploration. However, it lacks sufficient justification for its methodological design, clear explanations for key findings, and meaningful improvements in generative metrics.

---

> ### Author Response · Authors · 2025-11-26
> **Author Response 1/2**
>
> **1. This paper is an engineering work. The main contributions stem from combining existing components, e.g., 2D spatial convolutions in high-resolution stages, 3D causal attention, and increased latent channels—without introducing new conceptual insights.**
>
> 1. We respectfully disagree with the characterization that the work is merely an engineering combination of existing components. While our architecture leverages known primitive operations (e.g., Conv2D, Conv3D, causal attention), the core contribution of H3AE lies in how these components are systematically reorganized, profiled, and optimized to achieve an unprecedented combination of (i) ultra-high spatio-temporal compression, (ii) real-time mobile decoding, and (iii) preserved reconstruction fidelity, which prior work has not achieved. Existing video VAEs typically scale compression by sacrificing quality or introduce heavy modules that preclude mobile usage. In contrast, our design identifies and resolves bottlenecks by disentangling spatial and spatio-temporal processing and strategically distributing different block types to capture the spatial-temporal receptive field while maintaining efficiency. The design space is large and nontrivial, and our results show that only a very specific distribution of computation achieves the measured quality/speed/latency Pareto frontier.
> 2. Furthermore, the paper introduces Omni-objective training and latent consistency loss, which are novel to the video VAE domain and demonstrate consistent performance gains from the ablations.
> 3. Finally, our empirical results demonstrate large performance gains over state-of-the-art video VAEs, including LTX and Cosmos, even when those baselines operate at lower compression. The scalability, robustness across compression levels, and mobile deployability further indicate that this work contributes substantial new insights beyond engineering assembly.
>
> **2. The paper does not provide explanations for key findings, remaining at an empirical reporting level. For instance, it does not clarify why the proposed omni-objective training yields “free lunch” improvements instead of task conflict between t2v and I2V objectives.**
>
> We appreciate the reviewer’s interest in omni-objective training. We clarify that the “free lunch” effect of omni-objective training arises from the fact that T2V and I2V reconstruction are not competing tasks—both aim to reconstruct the same target video, with the I2V branch simply providing additional hierarchical features from the first frame. Injecting these features acts as a form of conditioning augmentation: it anchors early decoder features, reduces variance in feature alignment, and regularizes the reconstruction path without introducing contradictory gradients. Because the objective remains identical for both branches, there is no source of task conflict as in typical multi-task learning. Instead, the decoder becomes more robust to variations in latent codes, improving the plain T2V setting while further benefiting I2V (as in Fig. 5). We add the clarifications as highlighted in line#283 the revision.
>
> **3. While reconstruction metrics improve, generation results show little gain over baselines, suggesting limited impact on generative capability. Reconstruction and generation quality are not highly correlated; for generative tasks, improvements in generation metrics would be more meaningful.**
>
> We would like to clarify that the primary goal of this work is not to propose a new generative model or improve diffusion-based video generation quality, but to advance the design and training of video VAEs—a component whose impact is typically orthogonal to denoiser capacity. The generative results are included only as a sanity check to demonstrate that our highly compressed latent space retains sufficient “diffusability” for downstream use. Improving diffusion quality is outside the scope of this work and depends far more heavily on the denoiser architecture, training data curation and scale, and sampling strategies, rather than the VAE alone. Our contributions focus on achieving significantly higher compression, lower latency, and better reconstruction fidelity compared to prior VAEs, which represents the core technical novelty. As such, we encourage the reviewer to evaluate our work on its intended contributions: advancing video autoencoding efficiency and quality. The generation results simply confirm that our VAE imposes no issues or degradations on the diffusion model despite being efficient and performing aggressive latent compression.

---

> ### Author Response · Authors · 2025-11-26
> **Author Response 2/2**
>
> **4. It remains unexplained why the latent consistency loss is formulated as a KL divergence rather than a simpler distance such as MSE. Additional ablation studies are needed to justify this choice.**
>
> 1. We chose KL divergence because it directly reflects the true probabilistic structure of a variational autoencoder, where the encoder outputs a Gaussian posterior q() and the latent is explicitly sampled according to the mean and variance of the posterior (not deterministic ). Comparing two posteriors via MSE on means or samples would ignore the variance term and break the variational formulation. KL is the mathematically correct divergence between two Gaussians and naturally accounts for both the shift in means and the change in uncertainties.
>
> 2. Alternatively, we can use the entire VAE to reconstruct the fake sample instead of using fake posterior, in which case L1/L2 loss is applicable, and the method becomes ”pixel consistency loss”. Note that this setting further slows down training because of the decoder forward, while in our experiments, we find it offers no further gain compared to the encoder-KL setting. As a result, we stick to the latent consistency setting. Thanks for raising this discussion, we include this in the revision as highlighted in Appendix E.
>
> |        Loss        | PSNR  |  SSIM  |  rFVD |
> |:------------------:|:-----:|:------:|:-----:|
> |      Baseline      | 29.48 | 0.8280 | 147.1 |
> | Latent Consistency | 29.58 | 0.8299 | 142.2 |
> |  Pixel Consistency | 29.52 | 0.8286 | 143.9 |
>
> **5. The code is not released, which reduces transparency and reproducibility for a work that is primarily engineering-oriented.**
>
> Thank you for the comment. We fully agree with the importance of transparency and reproducibility. To support reproducibility, Appendix D provides the complete architectural specification, and we have added detailed training configurations and hyperparameters in the revised submission. Due to organizational policy constraints and considerations around data licensing and generative AI safety, unfortunately we are unable to release model weights at this time. However, our goal has been to make the work replicable despite this: the full architecture specification and training recipe together provide all necessary details for independent reproduction.

---

> > ### Comment · Reviewer_rERM · 2025-11-28
> >
> > Thanks for the authors’ reply, especially the clarification that the limited improvement in generation is due to the paper’s primary focus on compression rate. This addresses my concerns about the method’s performance, and I am therefore raising my score to 4.
> >
> > Why not higher? After reading other reviewers’ comments and the authors’ response, I still feel that the work mainly combines existing techniques to demonstrate the feasibility of such integration. In terms of new insights, for example, a deeper explanation of why T2V and I2V can mutually enhance each other, is lacking, offering only a general statement that they share the same objective. Hence, I am keeping my score at 4.

---

> > > ### Author Response · Authors · 2025-12-04
> > >
> > > We thank the reviewer for the reassessment and for raising the score. We would like to further clarify the terminology and provide a more in-depth explanation of the benefit of our omni-objective training.
> > >
> > > 1. Terminology clarification
> > >
> > > The terms “T2V” and “I2V” are from the scope of diffusion generator.
> > > From the VAE perspective, both modes are simply reconstruction, with the only difference being whether the decoder receives: (i) latent-only input (reconstruction purely from compressed representation), or (ii) latent + first-frame feature (reconstruction with a small amount of ground-truth leakage). So this is not multi-task learning, instead it is one VAE objective with or without an additional supervision signal.
> > >
> > > 2. Benefit of **with** additional first-frame signal (I2V usage mode)
> > >
> > > Leaking a ground-truth frame gives the decoder strong spatial anchors such as object contours and textures, which helps preserve structural fidelity under extreme compression. This is straightforward and consistent with common intuitions (I2V is a simpler task than T2V).
> > >
> > > 3. Why the benefit persists even when first-frame conditioning is **absent** (pure reconstruction)
> > >
> > > During omni-objective training, the decoder learns to produce accurate reconstructions both with and without the additional image guidance. This forces the latent representation and the hierarchical decoder features to become better aligned with the true data manifold: (i) The decoder cannot rely solely on the leaked frame (due to stochastic training ratio p). (ii) The latent pathway must remain informative and robust. (iii) Decoder feature statistics shift closer to the real video distribution, reducing drift.
> > > As a result, even when the first-frame condition is absent at inference time, the latent-only pathway inherits a more stable and better-aligned feature space, improving pure reconstruction performance as shown in the ablations.

---

### Official Review · Reviewer_uKBu · 2025-11-07

**Soundness:** 3
**Presentation:** 4
**Contribution:** 3
**Rating:** 8
**Confidence:** 4

**Summary:**

This paper introduces H3AE, a highly efficient and expressive autoencoder for video diffusion models that supports both text-to-video and image-to-video generation. The authors show that by distributing 2D and 3D computation in video VAEs, one can design architectures that are more efficient while maintaining reconstruction quality under stronger compression. The paper also studies training techniques for video VAEs and suggests several improvements that further stabilize training and improve both efficiency and quality. Finally, the proposed VAE is ablated against several strong baselines, and the authors demonstrate that competitive video diffusion models can be trained in the latent space of H3AE.

**Strengths:**

- As video diffusion models become increasingly common, improving their efficiency to enable real-time, on-device applications is an important direction. This paper helps bridge the gap between the efficiency of current video VAEs and the requirements of real-time processing on smaller devices.

- The idea of distributing 2D and 3D computation based on resolution is novel and interesting.

- The modified training techniques, such as the use of consistency loss and image conditioning, are both interesting and valuable to the broader community.

- The experiments are well designed and clearly show the improvements and contributions presented in the paper.

- The paper is well written and easy to follow.

**Weaknesses:**

- The paper would benefit from a more detailed explanation of why the proposed consistency loss can replace the GAN loss. From my experience, the GAN loss is typically essential for achieving sharp results. This finding is both interesting and surprising, and therefore might need further discussion.

- In Equation (1), it is unclear what the product operation represents. A brief explanation or notation clarification would improve readability.

- The paper contains a few minor typos that should be corrected in the final version.

**Questions:**

- It would be helpful to explain why the FVD score increases when using the GAN loss in Table 5, and how the proposed approach manages to avoid blurry outputs without it.

- In Equation (3), it appears that the model assumes the latent variables follow a Gaussian distribution. It would be helpful if the authors could clarify the motivation or theoretical justification for this assumption. Since the KL divergence weight is often set to a small value in practice, the latent distribution may deviate from a true Gaussian, so a brief discussion on how this affects the validity of the loss formulation would strengthen the paper.


- How does the model scale when latency is not a limiting factor? For instance, could larger architectures or higher-resolution settings further improve performance under relaxed efficiency constraints?

---

> ### Author Response · Authors · 2025-11-26
> **Author Response**
>
> **1. A more detailed explanation of why the proposed consistency loss can replace the GAN loss, …, and Q1: FVD score of GAN loss in Table 5, and how the proposed approach manages to avoid blurry outputs.**
>
> We thank the reviewer for highlighting this interesting aspect. We would like to clarify that our goal is not to replace GAN losses universally, but to show that in the context of high-compression video autoencoding, the proposed latent consistency loss is a more stable and task-aligned alternative. GAN loss acts in pixel space and often sharpens high frequency regions. However in the setting of high-compression video autoencoding, we find that GAN loss often brings dot or grid-like artifacts, as shown in Fig.3 and discussed in the Experiment section. Those artifacts can be reflected in quantitative metrics such as FVD. In contrast, LC loss regularizes at the distribution level by encouraging the encoder and decoder posteriors to agree, which helps preserve structural and temporal consistency while still improving perceptual sharpness. Importantly, LC loss does not preclude the use of GANs: as in the following table, we include an LC+GAN variant that further improves sharpness while maintaining more stable dynamics. We will expand the discussion in the paper and include this result.
>
> |    Loss    |  PSNR |  SSIM  |  LPIPS | FloLPIPS |  rFVD |
> |:----------:|:-----:|:------:|:------:|:--------:|:-----:|
> |  Baseline  | 29.48 | 0.8280 | 0.2627 |  0.2611  | 147.1 |
> |     GAN    | 29.35 | 0.8241 | 0.2631 |  0.2623  | 161.0 |
> |    Ours    | 29.58 | 0.8299 | 0.2541 |  0.2511  | 142.2 |
> | Ours + GAN | 29.60 | 0.8302 | 0.2519 |  0.2510  | 141.4 |
>
> **2. In Equation (1), it is unclear what the product operation represents. A brief explanation or notation clarification would improve readability.**
>
> Thanks for the suggestion, we improved the annotation of Equation 1 in the revision and highlighted the descriptions.
>
> **3. The paper contains a few minor typos that should be corrected in the final version.**
>
> Thanks for pointing out, we fix the typos as highlighted in the revision.
>
> **4. In Equation (3), it appears that the model assumes the latent variables follow a Gaussian distribution. It would be helpful if the authors could clarify the motivation or theoretical justification for this assumption. Since the KL divergence weight is often set to a small value in practice, the latent distribution may deviate from a true Gaussian.**
>
> Thanks for the insightful comment. In our VAE, the latent posterior $q(z|x)$ is explicitly parameterized as a Gaussian distribution with mean and variance predicted by the encoder, and sampling ($posterior.sample()$) is always performed through the reparameterization trick $z = \mu + \sigma \epsilon, \epsilon \sim \mathcal{N}(0,I)$. This parametric choice ensures that the posterior remains Gaussian by construction, regardless of the KL weight used during training. Empirically, we found that when using small KL regularization, the mean and variance of the posterior may deviate from $N(0,I)$, but still Gaussian. Therefore, computing the KL divergence between the “real” posterior and the “fake” posterior in Eq. (3) is mathematically well-defined and consistent with the underlying VAE parameterization. We add this discussion in line#315.
>
> **5. How does the model scale when latency is not a limiting factor? For instance, could larger architectures or higher-resolution settings further improve performance under relaxed efficiency constraints?**
>
> Thanks for the suggestion. We include a scaling study in the revision to show the generalization of H3AE. As shown in the following table, we relax the computation and memory constraint and scale up the network, and demonstrate good performance gains. In addition, we show that on higher resolution, H3AE consistently achieves higher PSNR and fidelity in zero-shot manner, which aligns with the behavior of popular video VAEs that pretraining on lower resolution but evaluating on higher resolution yields better reconstruction.
>
> | Resolution | #Params. (M) |  PSNR |  SSIM  |  rFVD |
> |:----------:|:------------:|:-----:|:------:|:-----:|
> |     512    |      197     | 30.23 | 0.8412 | 122.8 |
> |     512    |      401     | 30.93 | 0.8660 |  69.9 |
> |     768    |      197     | 30.75 | 0.8643 |  68.0 |

---

### Official Review · Reviewer_2ron · 2025-11-11

**Soundness:** 3
**Presentation:** 3
**Contribution:** 3
**Rating:** 8
**Confidence:** 3

**Summary:**

The authors introduce several improvements to VAE training and architecture that allow them to reach a new Pareto frontier on the reconstruction quality / compression ratio tradeoff. These improvements include:

- a new, computationally efficient architecture
- training the decoder to reconstruct the video both with and without being shown the first frame. They report that this improves decoder performance even when the first frame isn't shown.
- a "latent consistency loss" imposed on the decoder. While reconstruction loss encourages D(E(x)) to be close to x, latent consistency loss encourages E(D(E(x))) to be close to E(x). This improves reconstruction quality across all their metrics.

**Strengths:**

Very straightforward technology paper offering very straightforward improvements in VAE design and backing up those improvements with sensible empirical evidence. Well-written and presented.

**Weaknesses:**

No complaints. This paper is thorough and professional beyond my ability to find criticisms.

**Questions:**

N/a

---

> ### Author Response · Authors · 2025-11-26
> **Author Response**
>
> We sincerely thank the reviewer for the positive and encouraging feedback.

---

### Author Response · Authors · 2025-11-26
**General Response**

**We thank the reviewers and the Area Chair for their valuable comments and careful assessment of our submission**. We would like to take this opportunity to clarify the intended contributions of this work and to ensure the evaluation aligns with the core scope of our research.

1. This work establishes **the first systematic study of the Pareto frontier between video VAE compression, reconstruction quality, and decoding throughput**.
While we build upon standard architectural primitives, the central scientific contribution is **not** an engineering assembly, but a principled design space exploration that analyzes where computation should be allocated to achieve ultra-high compression without sacrificing fidelity or mobile feasibility. Prior VAEs either degrade quality significantly at these compression levels, or become impractical for deployment. Our methodology reveals previously unknown structural bottlenecks and architectural trade-offs, leading to a design that advances the efficiency–quality Pareto frontier as visualized in Fig. 1.

2. We introduce new learning objectives that advance video autoencoding beyond prior work.
The **latent consistency loss** and **omni-objective training** are novel contributions tailored to the challenges of high-compression video representation. They deliver stable training, reduce temporal artifacts, and produce measurable quality gains across resolutions and compression ratios. As reviewers note, these innovations carry significance beyond the specific architecture, and are expected to benefit the broader community working on video representation learning.

3. Clarifying the scope: **H3AE focuses on Video Autoencoding, not video diffusion**.
We acknowledge concerns raised about the comparisons and advances of generative quality. However, the diffusion experiments serve solely as a sanity check to confirm compatibility of H3AE with downstream generators. Improving diffusion-based generation quality requires additional research on denoiser design, data curation, and ODE sampling, which is orthogonal to (but not the primary focus) of H3AE. The primary evaluation criteria for this paper are autoencoding fidelity, temporal coherence, and efficiency, which we report comprehensively through PSNR/SSIM/rFVD and mobile latency measurements. We respectfully request reviewers focus on VAE-specific performance, which is the target contribution of this submission.

We have incorporated additional clarifications, ablations, comparisons, and architectural details in the revision. We hope our responses and revision helps reaffirm the novelty, rigor, and relevance of our contributions toward advancing efficient video representation learning. Thank you again for your time and consideration!

---

### Meta-Review · Area_Chair_k3Lz · 2026-01-06

**Summary:**

This paper presents H3AE, a video autoencoder designed to achieve ultra-high compression ratios with real-time mobile decoding capability while maintaining reconstruction quality. The work explores architectural design choices, proposes an omni-objective training strategy that enables a single VAE to support both T2V and I2V reconstruction, and introduces a latent consistency loss to improve perceptual quality.

The reviews are sharply divided: two reviewers strongly support acceptance (both scoring 8), recognizing the systematic design space exploration and substantial empirical improvements, while three reviewers express concerns about limited conceptual novelty and presentation issues (scoring 2, 4, and 4).

**Reviewer Concerns:**

Concerns Successfully Addressed:
- Clarification questions (uKBu, DffG): The authors adequately addressed technical clarifications about the latent consistency loss formulation, the product operation in Eq. 1, architectural details (kernel sizes, downsampling), and terminology (RoPE, QK-norm, "4ch"). Reviewer uKBu expressed satisfaction with the responses.
- GAN loss replacement (uKBu): The authors provided a compelling explanation with ablation showing that LC loss + GAN achieves the best results, and clarified that LC loss is not meant to universally replace GAN but provides a more stable alternative for high-compression video autoencoding.
- Scalability analysis (uKBu): The authors included a scaling study demonstrating performance gains with larger models and higher resolutions, addressing concerns about model behavior under relaxed constraints.
- Omni-objective training explanation (rERM): The authors provided detailed explanation of why T2V and I2V objectives don't conflict, framing it as conditioning augmentation rather than multi-task learning.

Outstanding Concerns:
- Limited conceptual novelty (rERM, WhDk, DffG): Despite rebuttal, reviewers rERM and DffG remain concerned that the work primarily combines existing components (2D/3D convolutions, causal attention) without introducing fundamentally new architectural insights. Reviewer rERM acknowledges "valuable engineering exploration" but maintains the score at 4, noting lack of "deeper explanation" for key findings.
- Fairness of comparisons (WhDk): Reviewer WhDk maintains that comparing models at different compression ratios in Table 1 makes it "difficult to really judge if techniques proposed in this paper really are much more beneficial." While authors argue they are advancing the Pareto frontier (visualized in Fig. 1), the reviewer's concern about direct comparison fairness was not fully resolved.
- Generation quality improvements (rERM): Despite understanding the VAE vs. diffusion distinction, Reviewer rERM notes that "reconstruction and generation quality are not highly correlated" and suggests that for practical impact, generation improvements would be more meaningful. This philosophical difference in evaluation criteria persists.

**Reviewer Scores:**

- Reviewer 2ron: Likely remains at 8/10.
- Reviewer uKBu: Likely remains at 8/10.
- Reviewer rERM: Confirmed at 4/10 after rebuttal (increased from 2).
- Reviewer WhDk: Likely remains at 4/10.
- Reviewer DffG: Likely remains at 2/10 or potentially 4/10.

---

### Decision · Program_Chairs · 2026-01-26

Reject